# Sugar-Sweetened Beverage Consumption in Adults: Evidence from a National Health Survey in Peru

**DOI:** 10.3390/nu14030582

**Published:** 2022-01-28

**Authors:** Wilmer Cristobal Guzman-Vilca, Edwin Arturo Yovera-Juarez, Carla Tarazona-Meza, Vanessa García-Larsen, Rodrigo M. Carrillo-Larco

**Affiliations:** 1School of Medicine “Alberto Hurtado”, Universidad Peruana Cayetano Heredia, Lima 15120, Peru; wilmer.guzman@upch.pe; 2CRONICAS Centre of Excellence in Chronic Diseases, Universidad Peruana Cayetano Heredia, Lima 15120, Peru; 3Sociedad Científica de Estudiantes de Medicina Cayetano Heredia (SOCEMCH), Universidad Peruana Cayetano Heredia, Lima 15120, Peru; 4Facultad de Medicina, Universidad Nacional de Trujillo, Trujillo 13007, Peru; earturo34@hotmail.com; 5Program in Human Nutrition, Department of International Health, The Johns Hopkins Bloomberg School of Public Health, Baltimore, MD 21205, USA; ctarazo1@jh.edu (C.T.-M.); vgla@jhu.edu (V.G.-L.); 6Center for Global Non-Communicable Diseases Research and Training, Johns Hopkins University, Baltimore, MD 21218, USA; 7Universidad Cientifica del Sur, Lima 15067, Peru; 8Department of Epidemiology and Biostatistics, School of Public Health, Imperial College London, London W2 1PG, UK; 9Universidad Continental, Lima 15046, Peru

**Keywords:** SSB, sugar-sweetened beverages, adults, Peru, Latin America, diabetes, obesity

## Abstract

High consumption of sugar-sweetened beverages (SSB) is associated with a high risk of non-communicable diseases. Evidence of SSB consumption is needed to inform SSB-related policies, especially in countries with a high consumption, such as Peru. Using data from Peru’s National Health Survey conducted in 2017–2018, the consumption of homemade and ready-to-drink SSB was estimated from a single 24 h dietary recall, accounting for socio-demographic and health-related variables. Regression models were fitted to assess which variables were linked to a high/low SSB consumption. There were 913 people and mean age was 37.7 years (95% confidence interval (CI): 36.9–38.6). Mean consumption (8 oz servings/day) of homemade SSB (1.2) doubled that of ready-to-drink SSB (0.5). The intake of homemade and ready-to-drink SSB was higher in men (1.3 and 0.7) than women (1.1 and 0.3). The intake of ready-to-drink SSB was higher in urban (0.6) compared to rural (0.2) populations. People aware of having diabetes had a lower consumption of both ready-to-drink (0.9 vs. 0.4) and homemade SSB (1.3 vs. 0.8) than those unaware of having diabetes. Male sex and living in urban locations were associated with higher ready-to-drink SSB intake. Older age was associated with a higher intake of homemade SSB. Amongst Peruvian adults, the consumption of SSB products (particularly homemade) remains high. Population-wide interventions should also aim to improve awareness of the nutritional components of homemade beverages.

## 1. Introduction

The frequent consumption of sugar-sweetened beverages (SSB) is linked to a higher prevalence and an increased risk of metabolic-mediated non-communicable diseases (NCDs), including non-alcoholic fatty liver disease, diabetes, and coronary heart disease, all of which are worsened by obesity and are important drivers of the global NCDs burden [1,2,3,4,5,6,7]. In 2019, ~240,000 deaths from NCDs were attributed to a diet high in SSB, with three quarters of these deaths occurring in low- and middle-income countries (LMICs) [4,8]. In response, the World Health Organization (WHO) has endorsed recommendations aimed at the population- (e.g., taxation on SSB) and individual-level (e.g., reducing free sugar consumption to < 10% of total energy intake) to reduce SSB consumption worldwide [9,10].

Global and regional efforts have been made to produce comparable estimates of SSB consumption at the national level [11,12]. International estimates from 2010 showed that the highest SSB consumption was observed in Latin America and the Caribbean (LAC). A recent regional analysis provided estimates of SSB consumption in eight LAC countries using data collected in 2014–2015. In this work, Argentina and Peru were the leading consumers of SSB, both with a mean intake of >900 cm^3^/day (>3.88 oz servings/day) [12]. Although this work studied a nationally representative sample, they only included people from urban areas [12], which could hide urban-rural differences on the SSB consumption in countries such as Peru [13]. Given the alarming increase in obesity and poor diet quality in LAC [14], it is critical to generate nationally representative evidence that contributes to illustrate and inform the current consumption of SSB in the region.

Using data from a National Health Survey in Peru, this study aimed at examining the consumption of ready-to-drink and homemade SSB in the adult population. The study explores differences in consumption by urban and rural settings and considers several diet-related NCDs to inform evidence-based interventions in Peru and other similar LAC countries.

## 2. Materials and Methods

### 2.1. Study Design

This is a cross-sectional analysis of a national survey conducted by the Peruvian National Centre for Food and Nutrition (CENAN, for its acronym in Spanish). This survey was conducted in 2017–2018 on a nationally representative sample of Peruvian adults [15]. CENAN’s survey included questions on dietary intake, anthropometrics measurements (height, weight and waist circumference (WC)), blood pressure, and blood biomarkers (fasting glucose), which were taken by trained field workers using standard protocols [15].

### 2.2. Study Population

Adults aged 18 to 59 years old, living in urban and rural areas, were included in the survey [15]. The current analysis was limited to survey participants with complete dietary intake data. No further exclusion criteria were applied.

### 2.3. Measuring Dietary Intake

Details on the dietary assessment have been reported elsewhere [15]. Briefly, all participants were invited to complete a 24 h recall questionnaire administered by trained nutritionists. During the 24 h recalls, people were asked to report all individual foods consumed (along with their ingredients) in the previous 24 h; when possible, food items and standard portion sizes were directly weighted to determine the quantity (in grams or milliliters) of each food and beverage consumed by the individual the day before. To ensure the inclusion of all days of the week, the day of the 24 h recall was randomly selected. All participants had at least one 24 h recall, and 20%, had two non-consecutive 24 h recalls. In this work, we only used the first 24 h recall. To further inform this decision, we used paired *t*-test to compare the mean SSB consumption between the first and second 24 h recall amongst those who had two valid 24 h dietary recalls (*n* = 139). There was no statistically significant difference in both the homemade (*p* = 0.6169) and ready-to-drink (*p* = 0.5575) mean SSB consumption between the two 24 h recalls (Appendix A).

### 2.4. Outcome

The primary outcome of interest was SSB consumption, estimated as a portion in 8 oz servings/day [11]. SSB was defined SSB as any liquids that are sweetened with various forms of added sugars [1]. Consistent with recent evidence from LAC [12,16], SSB were divided into two categories: homemade, which included homemade beverages with added sugar (e.g., tea, coffee, fruit juices), and ready-to-drink SSB, which included beverages with added sugar purchased ready to drink (e.g., soda, energy drinks, sweetened water, nectar). If ready-to-drink SSB were mixed with alcoholic beverages (e.g., coke mixed with vodka), we only considered the amount of the SSB (i.e., the amount of the alcoholic beverage was removed). Pure milk consumption was not included as it is likely to have a different consumption pattern and nutritional content in Peru [11]; nonetheless, those drinks mixed with milk were included if they included sugar (e.g., coffee with milk with added sugar). Additionally, artificially sweetened beverages (e.g., stevia) were excluded.

Regarding homemade SSB, we restricted the analysis to homemade beverages that explicitly reported sugar or any form of added sugar (e.g., honey) as an ingredient [1]. For example, when sugar was explicitly reported as an ingredient of lemonade, this beverage was considered a homemade SSB. Conversely, if sugar was not explicitly reported as an ingredient of lemonade, this beverage was not considered a homemade SSB. Nonetheless, because some individual servings of homemade SSB may have not reported sugar (i.e., underreporting), we also performed a sensitivity analysis for homemade SSB consumption in which all servings of fruit juice, coffee, and tea were considered as homemade SSB regardless of whether the sugar content was explicitly reported. We did not consider a lower limit of sugar content per serving for homemade SSB. As long as a homemade beverage had added sugar (regardless of the amount), it was considered a homemade SSB.

### 2.5. Covariates

We stratified our outcomes of interest by the following demographic variables: sex, 10-year age groups and urban/rural location. We also stratified the outcome variable by the relevant correlates: educational level, body mass index (BMI), abdominal obesity, diabetes, and hypertension status.

Educational level was originally defined on the survey based on the highest level of schooling reported by the participant. There were the following options: no or less than primary education, primary education, secondary education, higher education, and post-graduate education. In the paper, we reported the level of education in three groups: low (no or less than primary/primary education), medium (secondary education), and high (higher/post-graduate education). No additional socio-economic variables were available.

BMI (kg/m^2^) was computed using measured weight and height. We excluded BMI records outside the range 10–80 kg/m^2^, which were deemed biologically implausible. For descriptive purpose, we categorized BMI into normal (BMI < 25 kg/m^2^), overweight (BMI 25–29.9 kg/m^2^), and obesity (BMI ≥ 30 kg/m^2^).

Abdominal obesity was defined using measured waist circumference (WC). We classified individuals into two groups based on WC: no abdominal obesity (WC < 90 cm in men and < 80 cm in women), and abdominal obesity (WC ≥ 90 cm in men and ≥ 80 cm in women) [17].

Although the survey included three blood pressure measurements, the third blood pressure measurement was taken in < 2% of participants. We used the second blood pressure measurement only (i.e., the first and third blood pressure records were discarded). Of note, there were no substantial differences between the first and second blood pressure measurements (Appendix A). Self-reported information on hypertension diagnosis and antihypertensive treatment were obtained using the following questions: “have you ever been told by a physician or another healthcare worker that you have high blood pressure or hypertension?”, and “In the last two weeks, have you received any drug to treat hypertension that has been prescribed by a physician?” For descriptive purposes, we defined hypertension as those who had blood pressure ≥ 140/90 mmHg, or answered yes to the self-reported hypertension diagnosis question, or yes to the antihypertensive treatment question. We defined people aware of having hypertension (i.e., self-reported hypertension) as those who answered yes to the self-reported hypertension diagnosis question or yes to the antihypertensive treatment question. Those unaware of having hypertension were defined as having blood pressure ≥ 140/90 mmHg and answered no to the self-reported hypertension diagnosis and treatment questions. The otherwise healthy group (i.e., not with hypertension) were those with blood pressure < 140/90 mmHg and answered no to the self-reported hypertension diagnosis and treatment questions.

Type 2 diabetes mellitus (T2DM) status was assessed with measured fasting plasma glucose [15], along with self-reported information on diabetes diagnosis and treatment: “have you ever been told by a physician or another healthcare worker that you have high blood sugar or diabetes?” and “in the last two weeks, have you received any drug to treat diabetes that has been prescribed by a physician?”. We defined people with diabetes as those who had a fasting plasma glucose ≥ 126 mg/dL or answered yes to the self-reported diagnosis question or yes to the treatment question. We defined people aware of having diabetes (i.e., self-reported T2DM) as those who answered yes to the self-reported diagnosis question or yes to the treatment question. We defined people unaware of having T2DM as those who had a fasting plasma glucose ≥ 126 mg/dL and answered no to the self-reported diagnosis or treatment questions. The otherwise healthy group (i.e., not with diabetes) included those who had a fasting plasma glucose < 126 mg/dL and answered no to the self-reported diagnosis or treatment questions.

### 2.6. Statistical Analyses

To estimate the daily homemade and ready-to-drink SSB consumption, we added the quantities of all homemade or ready-to-drink SSB servings reported in a single 24 h recall per individual. Afterwards, SSB consumption data (in ml) was converted into ounces multiplying it by 0.033814, and we estimated the number of servings (8 oz) per day per individual.

Population characteristics along with their 95% confidence interval (95% confidence interval (CI)) were described accounting for the complex survey design [15]. In order to assess which variables were associated with a higher or lower SSB consumption in 8 oz servings/day, we fitted a linear regression model using the svyglm command in R, where the association was considered statistically significant at a *p* < 0.05. The regression model accounted for the complex survey design and included crude and adjusted analyses (adjusted for sex, age, urban/rural location, educational level, BMI category, abdominal obesity, diabetes and hypertension status). The outcome in the model was SSB consumption in 8 oz servings/day. We reported the results of the adjusted regression model on the main text, and the results of the crude regression models are available as Appendix A. All analyses were conducted in R (version 4.0.3, R Core Team, Vienna, Austria). The analysis code and datasets are also available as Appendix A.

## 3. Results

There were 913 participants with 24 h recall data (Table 1). The mean age was 37.7 years (95% CI: 36.9–38.6) and 42.0% of the population were men. The prevalence of overweight was 37.5% (95% CI: 33.8–41.4%) and the prevalence of obesity was 27.5% (95% CI: 24.1–31.1%). The prevalence of abdominal obesity was 71.4% (95% CI: 67.9–74.4%). The proportion of people with diabetes was 14.0% (95% CI: 11.4–17.1%), and those with self-reported diabetes represented 4.6% (95% CI: 3.2–6.5%). The prevalence of hypertension was 9.9% (95% CI: 7.8–12.4%), and the prevalence of unaware hypertension was 2.5% (95% CI: 1.6–3.9%).

### 3.1. National Estimates of SSB Consumption According to 10-Year Age Groups and Sex

The mean consumption of ready-to-drink SSB was half as that of homemade SSB: 0.5 (95% CI: 0.4–0.6) vs. 1.2 (95% CI: 1.0–1.3). The mean consumption of homemade SSB was higher in men (1.3; 95% CI: 1.1–1.5) than women (1.1; 95% CI:0.9–1.2). The mean consumption of ready-to-drink SSB in men (0.7; 95% CI: 0.6–0.9) was more than double that of women (0.3; 95% CI: 0.3–0.4) (Table 2).

Overall, the mean consumption of ready-to-drink SSB was similar (≤0.5 servings/day) between age groups (Table 2). Stratified by sex, younger age groups had a slightly higher mean consumption of ready-to-drink SSB in men (Appendix A). Conversely, the mean consumption of homemade SSB was higher in older age groups; for example, mean consumption of homemade SSB in those aged 50–59 years (1.6; 95% CI: 1.3–1.9) almost doubled that of 18–29 years (0.9; 95% CI: 0.8–1.1) (Table 3).

### 3.2. National Estimates of SSB Consumption According to Urban/Rural Location

There was a three-fold difference in the mean consumption of ready-to-drink SSB between people living in urban locations (0.6; 95% CI: 0.5–0.7) than those living in rural locations (0.2 (95% CI: 0.1–0.2) (Table 2). The mean consumption of homemade SSB was similar between those living in urban (1.2; 95% CI: 1.0–1.5) and rural (1.1; 95% CI: 1.0–1.3) locations (Table 3).

### 3.3. National Estimates of SSB Consumption According to Educational Level

The mean ready-to-drink SSB consumption was the highest amongst those with medium level of education (0.7; 95% CI: 0.5–0.8) and lowest amongst those with a low level of education (0.2; 95% CI: 0.1–0.3) (Table 2). There was no substantial difference in the mean homemade SSB consumption across educational levels (Table 3, Appendix A).

### 3.4. National Estimates of SSB Consumption According to BMI Categories and Abdominal Obesity

Regarding ready-to-drink SSB, we observed a slightly positive trend with higher BMI; that is, the mean consumption of ready-to-drink SSB was slightly higher in those with obesity compared to those with normal weight (Table 2). The opposite pattern was observed for homemade SSB (Table 3): the mean consumption of homemade SSB tended to be slightly lower from normal weight and overweight to obesity.

The mean consumption of ready-to-drink SSB was virtually the same in people with and without abdominal obesity (Table 2), but the mean consumption of homemade SSB was slightly lower in those with abdominal obesity (Table 3).

### 3.5. National Estimates of SSB Consumption According to Diabetes and Hypertension Status

Regarding diabetes status, the highest mean consumption of ready-to-drink and homemade SSB was observed in those with unaware diabetes (Table 2 and Table 3). The lowest mean consumption was observed in those with self-reported diabetes and the otherwise healthy population regarding ready-to-drink SSB, and in those with self-reported diabetes regarding homemade SSB. The mean consumption of both homemade and ready-to-drink SSB was virtually the same across the hypertension groups (Table 2 and Table 3).

### 3.6. Regional Estimates of SSB Consumption

The mean homemade SSB consumption was higher than ready-to-drink SSB consumption across all regions except two (Tacna and Madre de Dios) (Appendix A). The regions with the highest mean consumption of homemade SSB were mainly in the Highlands (Junin, Pasco and Cajamarca). Conversely, the regions with the highest consumption of ready-to-drink SSB were mainly in the Coast (Tacna, Ica and Callao), whereas the regions with the lowest consumption of ready-to-drink SSB were mainly in the Highlands (Huancavelica, Ayacucho and Amazonas).

### 3.7. Factors Associated with a Higher or Lower SSB Consumption

Regarding ready-to-drink SSB consumption, female sex (*p* = 0.0006) and having a low educational level (*p* = 0.0038) were both associated with an average consumption of 0.3 less servings than being male and having a medium educational level, respectively. Living in urban locations (*p* < 0.0001) was associated with an average consumption of 0.4 more servings than living in rural locations. There were two associations borderline significant. The average consumption of ready-to-drink SSB decreased 0.01 servings with higher age (*p* = 0.0830) and having a high educational level was associated with 0.2 less servings than having a medium educational level (*p* = 0.0620). Associations between ready-to-drink SSB consumption and other variables were not statistically significant (Appendix A).

Regarding homemade SSB consumption, the mean consumption increased 0.03 servings with higher ages (*p* < 0.0001). Compared to the medium educational level, the low educational level was associated with 0.4 less servings (*p* = 0.0188), whereas having a high educational level was associated with 0.2 more servings (*p* = 0.0861). Central obesity was associated with an average consumption of 0.5 less servings compared with no central obesity (*p* = 0.0064). Associations between homemade SSB consumption and other variables were not statistically significant (Appendix A).

### 3.8. Sensitivity Analysis for Homemade SSB: Homemade SSB Regardless of Whether Sugar Content Was Explicitly Reported

The mean consumption of homemade SSB was substantially higher in the sensitivity analysis compared with the main analysis. For example, the mean consumption of homemade SSB in men was 3.8 (95% CI: 3.4–4.1) in the sensitivity analysis vs. 1.3 (95% CI: 1.1–1.5) in the main analysis. Not consistent with the main analysis, there was no clear pattern according to age groups (Appendix A) in homemade SSB consumption. Not consistent with the main analysis, the mean homemade SSB consumption was slightly higher in those with a medium level of education compared to the other educational levels (Appendix A). Consistent with the main analysis, there were no urban-rural differences in homemade SSB consumption (Appendix A). Furthermore, obese people had a slightly lower homemade SSB consumption compared to those with a normal weight (Appendix A), as it was the case in the main analysis. Those unaware of having diabetes and hypertension had a higher consumption of homemade SSB than those aware of having diabetes (Appendix A) and hypertension (Appendix A), respectively. Of note, the diabetes-related finding is consistent with the main analysis, but not the hypertension-related finding. Contrasting with the main analysis, the regions with the highest homemade SSB consumption in the sensitivity analysis were in the Coast (Appendix A).

## 4. Discussion

### 4.1. Main Findings

In this nationally representative survey of Peruvian adults, we investigated recent estimates of the SSB consumption profile in Peru. We found that the consumption of homemade SSB is higher than that of ready-to-drink SSB, regardless of sex, age, urban/rural location, and educational level. At the national level, we observed that men consumed more homemade and ready-to-drink SSB than women. When we stratified by age groups, the mean consumption of homemade SSB was higher in younger than older age groups. In men, those in the younger age groups had the highest ready-to-drink SSB consumption. We also observed that people living in urban areas had a higher consumption of ready-to-drink SSB than those living in rural areas. Regarding educational levels, people with a medium educational level had the highest ready-to-drink SSB consumption, whereas those with a low educational level had the lowest ready-to-drink SSB consumption. Although people with obesity had a higher consumption of ready-to-drink SSB, they had lower consumption of homemade SSB compared to those with normal weight. People aware of having diabetes had a lower consumption of both ready-to-drink and homemade SSB than those unaware of having diabetes. These results highlight the need for interventions to reduce the consumption of homemade SSB in Peru, and to also reduce the consumption of ready-to-drink SSB in urban areas; also, the results pinpointed that people with high SSB consumption are unaware they may have NCDs (e.g., diabetes).

### 4.2. Research in Context

A global work provided comparable estimates of mean consumption of SSB across 187 countries in 1990 and 2010 stratified by age and sex [11]. Although the results were reported for Peru, they did not use any data source from Peru (i.e., results for Peru were modelled). Therefore, we advanced these results by leveraging on nationally representative data, reporting recent estimates (2018), and reporting SSB consumption stratified by groups with relevant policy implications: homemade (i.e., not potentially taxed) and ready-to-drink (i.e., potentially taxed).

Compared to the global results, our estimates of mean consumption of SSB were slightly higher in both men and women [11]. In men, the mean (total) SSB consumption in Peru by Singh et al. [11] was <1.5 servings/day, whereas our estimates in men were 1.3 and 0.7 servings/day of homemade and ready-to-drink SSB, respectively. In women, Singh et al. [11] estimated a mean consumption of <1.4 servings/day, whereas our estimates in women were 1.1 and 0.3 servings/day of homemade and ready-to-drink SSB, respectively. These disparities could be explained by several factors. First, while they provided estimates for people aged ≥ 20 years [11], we studied people aged 18–59 years, which may have a higher SSB consumption; had we studied older populations, our estimates could have diluted. Second, we provided results using data collected in 2017–2018, whereas the results by Singh et al. were modelled for 2010 [11]; therefore, the differences could be attributed to an increase in SSB consumption since 2010.

A regional work provided estimates of mean consumption of both homemade and ready-to-drink SSB across eight LAC countries in 2014–2015; 24 h recalls were conducted to measure dietary intake in urban samples representative at the national level [12]. Notably, their estimates for Peru for ready-to-drink SSB were fairly similar to our estimates in milliliters, but their estimates for homemade SSB were slightly higher than ours. For example, the mean consumption of homemade SSB in men was 835.9 mL/day by Kovalskys et al. [12] vs. 306.5 mL/day by our estimates; in women, they reported 685.9 mL/day [12] vs. 250.9 mL/day by our estimates. The difference with our estimates could be explained by different dietary assessments: we focused only on individual beverages that reported sugar as an ingredient as our data source collected information about food servings along with their ingredients. Even though Kovalskys et al. [12] collected data on food servings, data on added sugar was estimated through specialized tools (Nutrition Data System for Research software, University of Minnesota Nutrition Coordinating Center, Minneapolis, MN, USA) [18]. Another potential explanation could be because we analyzed one 24 h dietary recall, whilst Kovalskys et al. [12] leveraged on two 24 h dietary recalls.

We reported that, when stratified by diabetes status, the mean consumption of both homemade and ready-to-drink SSB was the highest amongst those with unaware diabetes. This finding is consistent with a previous study in the U.S. [19] and advances the literature, which only focused on self-reported diabetes [20,21]. We found higher SSB intake in people who did not know they had diabetes compared to those with self-reported diabetes. This may suggest that people with diagnosed diabetes tend to be more aware of their health status and may follow healthy diets, including less SSBs consumption. This may be a consequence of tailored educational programs [22], for example those delivered at hospitals or specialized diabetes clinics [23,24]. Additionally, this may be a consequence of mass media campaigns whereby messages to promote healthy diets are delivered, specially targeting people with comorbidities, such as diabetes [25,26]. Our results suggest that these campaigns, either personalized or in mass media, should also include recommendations about homemade SSBs. This finding also calls to strengthen diabetes diagnoses, as almost half of the diabetes population is undiagnosed in LAC countries [27,28].

### 4.3. Strengths and Limitations

We provided the first nationally representative estimates of the mean consumption of both homemade and ready-to-drink SSB in Peru. Furthermore, we delivered this evidence stratified by relevant socio-demographic and clinical variables; additionally, we reported our results in 8 oz servings/day, which is comparable to global and regional works [11,12].

Nonetheless, our study has limitations. First, consumption estimates were derived from a single 24 h recall, which could have underestimated our results. However, as the use of a single 24 h recall underestimates the consumption of infrequent meals and SSB consumption appeared to be already high in Peru [11,12], thus our results should not have been greatly underestimated. Second, food reporting during the 24 h recalls could have been biased by social desirability or forgetfulness [29], underestimating SSB consumption. If our results are underestimated, then they may alert to a higher real SSB consumption in Peru. Finally, although we analyzed recent data (2017–2018), the events of recent years, the COVID-19 pandemic in particular, could have changed our findings. The COVID-19 pandemic could have increased SSBs consumption because people had to stay at home, having more chances to consume more homemade SSBs. Conversely, the COVID-19 pandemic could have decreased the overall SSBs consumption [30,31,32]. This may be the case, because the pandemic made the general population aware of unhealthy lifestyles and cardiometabolic diseases (e.g., diabetes) closely linked with COVID-19 severity. Both of these hypotheses are largely speculative and deserve further research with recent data. Future studies (e.g., before-and-after studies) should be conducted to provide current evidence about the changes in SSB consumption patterns during the COVID-19 pandemic in LAC.

### 4.4. Public Health Implications

From a public health perspective, the consumption of ready-to-drink SSB could be reduced or discouraged with population-based policies, such as taxes [33] or food labeling [34,35,36]. Peru has implemented both food labeling, as octagons in the package, and SSB taxation [37,38], and these policies were first introduced in 2019 and 1999, respectively [39]. The evaluation of these policies on food purchase and consumption is still ongoing. Conversely, homemade SSB would be less susceptible to these population-wide policies. Educational interventions oriented to a household level could represent an alternative, though more research is needed on this subject [40]. Mass media campaigns, using social media for young people, could reach large populations with high homemade SSB consumption. Likewise, community-based interventions and those with community health workers could provide valuable tools, though to the best of our knowledge these interventions have not been tested to reduce SSB consumption in adults.

## 5. Conclusions

In Peru, the consumption of homemade SSB was higher than the consumption of ready-to-drink SSB regardless of sex, age, urban/rural location, and educational level. This profile can contribute to inform policymakers and public health researchers to target groups of adults who have a higher consumption of SSB and might be more susceptible to SSB-related NCDs. Population-wide interventions, such as taxes and food labels, would have low effect on homemade SSB. Interventions would be needed to improve awareness of the nutritional components of homemade foods and beverages.

## Figures and Tables

**Table 1 nutrients-14-00582-t001:** Weighted distribution of demographic and clinical variables in the study population, overall and by sex.

Characteristics (95% CI, %, Unless Stated Otherwise)	Total (*n* = 913)	Men (*n* = 386)	Women (*n* = 527)
Total age (mean, years)	37.7 (36.9–38.6)	37.1 (35.7–38.5)	38.2 (37.1–39.3)
18–29 years	29.5 (26.4–32.9)	33.2 (27.9–39.1)	26.9 (22.8–31.4)
30–39 years	26.2 (22.9–29.8)	25 (20.4–30.3)	27.1 (22.8–31.8)
40–49 years	24.3 (21.2–27.7)	20.5 (16.5–25.1)	27.1 (22.8–31.8)
50–59 years	19.9 (17.1–23.1)	21.3 (17.1–26.3)	18.9 (15.5–22.9)
Body mass index (mean, kg/m^2^)	27.4 (27–27.8)	26.7 (26.1–27.3)	27.9 (27.4–28.5)
Proportion (%) with normal weight	35 (31.6–38.6)	39.1 (33.6–45)	32 (27.6–36.8)
Proportion (%) with overweight	37.5 (33.8–41.4)	38 (32.5–43.9)	37.2 (32.4–42.2)
Proportion (%) with obesity	27.5 (24.1–31.1)	22.9 (18.2–28.3)	30.8 (26.2–35.8)
Waist circumference (mean, cm)	92 (91–92.9)	92.7 (91.1–94.3)	91.4 (90.2–92.6)
Proportion (%) with central obesity	71.4 (67.9–74.7)	57 (51.3–62.5)	82.4 (78.4–85.8)
Proportion (%) with diabetes	14 (11.4–17.1)	15.1 (11.2–20.1)	13.2 (10–17.2)
Proportion (%) with self-reported diabetes	4.6 (3.2–6.5)	3.7 (2–6.9)	5.2 (3.4–7.8)
Proportion (%) unaware of having diabetes	8.9 (6.8–11.4)	11 (7.8–15.3)	7.3 (5–10.6)
Proportion of people with hypertension	9.9 (7.8–12.4)	12.2 (8.9–16.5)	8.1 (6–11)
Proportion of people with unaware hypertension	2.5 (1.6–3.9)	3.8 (2.1–6.8)	1.4 (0.7–2.8)

CI—confidence interval.

**Table 2 nutrients-14-00582-t002:** Ready-to-drink SSB consumption by socio-demographic and health-related variables.

Variable	Options for Variable	Number of Observations	Ready-to-Drink SSB Consumption (8 oz. Servings/Day) (Mean and 95% CI)	*p* value *
Sex	Men	386	0.71 (0.57–0.86)	<0.05
	Women	527	0.35 (0.26–0.43)	
Age group	18–29	258	0.55 (0.42–0.68)	0.15
	30–39	239	0.55 (0.37–0.73)	
	40–49	226	0.47 (0.33–0.62)	
	50–59	190	0.41 (0.26–0.55)	
Urban or rural	Rural	326	0.17 (0.10–0.24)	<0.05
	Urban	587	0.59 (0.49–0.69)	
Educational level	Low	227	0.17 (0.08–0.25)	<0.05
	Medium	339	0.68 (0.53–0.82)	
	High	346	0.50 (0.38–0.63)	
Body mass index category	Normal	329	0.41 (0.29–0.53)	<0.05
	Overweight	341	0.47 (0.36-0.59)	
	Obesity	234	0.66 (0.48–0.84)	
Abdominal obesity	No	278	0.50 (0.36–0.64)	0.84
	Yes	610	0.52 (0.42–0.62)	
Diabetes status	Not with diabetes	395	0.49 (0.41–0.57)	<0.05 †
	Total diabetes	518	0.69 (0.40–0.98)	
	Self-reported diabetes	40	0.40 (0.12–0.67)	
	Unaware diabetes	477	0.86 (0.43–1.28)	
Hypertension status	Not with hypertension	788	0.51 (0.42–0.60)	0.63 ‡
	Total hypertension	96	0.49 (0.30–0.68)	
	Self-reported hypertension	71	0.47 (0.25–0.70)	
	Unaware hypertension	25	0.55 (0.19–0.91)	

* For dichotomous variables (sex, urban/rural, abdominal obesity), *p* value from unpaired *t*-test. For the remaining variables except diabetes and hypertension status, *p* value for ANOVA test. † *p* value for *t*-test between total diabetes and not with diabetes. ‡ *p* value for *t*-test between total hypertension and not with hypertension.

**Table 3 nutrients-14-00582-t003:** Homemade SSB consumption by socio-demographic and health-related variables.

Variable	Options for Variable	Number of Observations	Homemade SSB Consumption (8 oz. Servings/Day) (Mean and 95% CI)	*p* value *
Sex	Men	386	1.30 (1.12–1.47)	<0.05
	Women	527	1.06 (0.93–1.19)	
Age group	18–29	258	0.93 (0.76–1.10)	<0.05
	30–39	239	1.08 (0.89–1.28)	
	40–49	226	1.18 (0.95–1.42)	
	50–59	190	1.57 (1.27–1.86)	
Urban or rural	Rural	326	1.24 (1.03–1.46)	0.32
	Urban	587	1.14 (1.01–1.27)	
Educational level	Low	227	0.97 (0.75–1.20)	0.14
	Medium	339	1.15 (0.98–1.32)	
	High	346	1.25 (1.07–1.43)	
Body mass index category	Normal	329	1.26 (1.05–1.47)	0.17
	Overweight	341	1.21 (1.02–1.40)	
	Obesity	234	0.97 (0.79–1.15)	
Abdominal obesity	No	278	1.32 (1.11–1.54)	<0.05
	Yes	610	1.12 (0.98–1.25)	
Diabetes status	Not with diabetes	395	1.18 (1.05–1.30)	0.67 †
	Total diabetes	518	1.09 (0.77–1.42)	
	Self-reported diabetes	40	0.76 (0.39–1.13)	
	Unaware diabetes	477	1.29 (0.84–1.74)	
Hypertension status	Not with hypertension	788	1.16 (1.04–1.28)	0.63 ‡
	Total hypertension	96	1.13 (0.83–1.42)	
	Self-reported hypertension	71	1.14 (0.79–1.50)	
	Unaware hypertension	25	1.07 (0.60–1.54)	

* For dichotomous variables (sex, urban/rural, abdominal obesity), *p* value for unpaired *t*-test. For the remaining variables except diabetes and hypertension status, *p* value for ANOVA test. † *p* value for *t*-test between total diabetes and not with diabetes. ‡ *p* value for *t*-test between total hypertension and not with hypertension.

## Data Availability

Datasets and analysis code in Appendix A.

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
