# Peer review of "Sugar-Sweetened Beverage Consumption in Adults: Evidence from a National Health Survey in Peru"

_nutrients, 2022, doi:10.3390/nu14030582_

Round 1
Reviewer 1 Report
This is a well conducted study that aims to understand adult consumption of homemade and ready-to-drink SSB in Peru, where excessive SSB consumption is great public health issue. This study contributes valuable data to the literature around the differences in consumption by rural and urban populations in Peru. Interestingly they also looked at how SSB intake varied by the non-communicable diseases that are generally linked with high consumption.
Some strengths include the use of 24 hour recalls to collect dietary data, their large and representative sample size, and their statistical analysis were appropriate and well done.
Some of minor comments I have are around the homemade SSB. I think the authors could provide more clarification around this.
- It is interesting the comparison of homemade to ready-to-drink SSBs. I understand that these drinks have different tax implications, which is a popular public health strategy to address SSB intake. In a sense, they are not comparable because of the variation in sugar added to homemade drinks. While the intake of these could be higher, does that necessarily mean more intake of sugar? This could be resolved with some detail around what was considered a homemade SSB (e.g. was there a lower limit of sugar content that would eliminate the drink from the SSB category?).
- The authors also make a statement line 99 that they only considered drinks that reported “sugar” as an ingredient. Yet in line 91 they cite literature that states SSB is any form of sugar that includes syrups, honey etc. Did they consider these drinks too?
- What about drinks made from drink mixes? Were these considered homemade or ready-to-drink?
Author Response
Baltimore, January 13th 2022
Ref.: Rebuttal letter “Sugar-Sweetened Beverages consumption in adults: Evidence from a National Health Survey in Peru” – Manuscript ID: nutrients-1564136 by Wilmer Cristobal Guzman-Vilca et al.
Nutrients
We are very grateful for the positive feedback and comments to our manuscript, and hereby include a point-by-point reply to the reports from the reviewer. Please, see the attachment.

Reviewer 2 Report
The paper “Sugar-Sweetened Beverages consumption in adults: Evidence from a National Health Survey in Peru" by Guzman-Vilca et al. is a cross-sectional study with the aim of assessing sugar-sweetened beverages consumption in Peruvian population.
The article is well written. The study has a good design. The article is logically divided into sections and subsections. There are several tables of good quality. The references cited are relevant and adequate. The work has an average degree of novelty and of good interest to the readers.
Comments:
- Introduction: the role of Sugar-Sweetened Beverages is not only relevant in the development of diabetes, coronary heart disease and obesity state, but it is also important in the development of non-alcoholic fatty liver disease (DOI: 3390/pr9010135), which in turn can increase the prevalence of cardiovascular disease onset (DOI: 10.31083/J.RCM2203082).
- Line 294-296 and 336-338: in diabetic patients it was reported a reduced intake of Sugar-Sweetened Beverages. This evidence maybe due too several issues: to the nutritional training, information, and increased worry about their health as they are ill. This may suggest a possible role of television nutritional educational programmes to provide tips to prevent disease onset and to increase basic nutritional knowledge.
- Line 355-356: it was reported that with COVID19 outbreak and due to most restrictions, world general population showed an increased prevalence of weight gain. On the other hand, it is possible that nutritional habits could have changed with increased COVID19 knowledge, as the definitions of newer risk factors for in-hospital admission and mortality, could have made aware the general population to avoid certain habits. In particular, as in point 1, the importance of the liver also in COVID19 prognosis, could have affected Sugar-Sweetened Beverages and also other foods (DOI: 10.1371/journal.pone.0243700).
Author Response

(The authors gave the same response as above.)
